# Barriers and facilitators to physical activity: A comparative analysis of transplant athletes competing in high intensity sporting events with other transplant recipients

**Bart Rienties[1], Elaine Duncan[2]\*, Perry Judd[3], Ben Oakley[4], Liset H. M. Pengel[5], Keetie Roelen[6], Nicholas Topley[7]**

**1** The Institute of Educational Technology, The Open University, Milton Keynes, United Kingdom, **2** Department of Psychology, School of Health and Life Sciences, Glasgow Caledonian University, Glasgow, United Kingdom, **3** The Prince Charles Hospital, Brisbane, Australia, **4** School of Education, Childhood, Youth & Sport, The Open University, Milton Keynes, United Kingdom, **5** Erasmus MC Transplant Institute, University Medical Centre Rotterdam, Rotterdam, The Netherlands, **6** Centre for the Study of Global Development, The Open University, Milton Keynes, United Kingdom, **7** School of Medicine, Cardiff University, Cardiff, United Kingdom

\* edu@gcu.ac.uk

**Data Availability Statement:** Interview data cannot be shared publicly because of the interviews contain very sensitive and private information

## Abstract

### Background

There is widespread recognition that many transplant recipients struggle to become and remain physically active. However, some transplant recipients do undertake strenuous training and significant physical activity (PA) and participate in intensive sports.

### Aim

This study sought to understand facilitators and barriers to be physically active for Transplant Athletes (TXA) compared to a group of Dutch transplantees. This explorative mixed methods study analysed race performance and interview data from TxA who participated in cycling and/or the sprint triathlon at the World Transplant Games 2023, and compared their lived experiences in terms of barriers and facilitators of PA with those of 16 transplantees in a study from the Netherlands previously published in this journal.

### Methods

Using Patient and Public Involvement and engagement (PPI), race data from World Transplant Games 2023 and subsequent in-depth interviews were used from 27 TxA. A visual artefact of barriers and facilitators from the previous Dutch study was used to prompt identification and discussion of barriers and facilitators of PA. Interview data were coded by three coders.

about participants' medical, physical and mental transitions before/during/after transplantation. However, the anonymised quantitative data of the reported barriers and facilitators to Physical exercise per participant are available at the Open University Institutional Data (ORDO) at https://doi.org/10.21954/ou.rd.c.7279993.v1 Access for researchers who meet the criteria for access to confidential data, including redacted transcripts from three interviews.

**Funding:** The author(s) received no specific funding for this work.

**Competing interests:** The authors have declared that no competing interests exist

## Results

Many of the barriers to PA previously reported by Dutch transplant recipients were not shared by the majority of TxA in this study. The TxA in this study reported significantly lower physical limitations, lower fear to undertake exercise, and no comorbidity issues for TxA. Furthermore, TxA perceived they received substantial social support, had the strength to do PA, and were in control of their weight.

## Conclusion

Several TxA reported a lack of understanding from medical and other professionals about the appropriate intensity of PA. An evidence-based framework of PA for transplant recipients and transplant athletes is needed for safe and appropriate PA.

## Introduction

Solid organ transplant surgery (e.g., heart, kidney, lung, liver, pancreas) and hematopoietic cell transplants (e.g., bone marrow) are life-saving medical procedures. Annually around 130,000 transplant surgeries take place globally [1]. After a successful transplantation, recipients are encouraged to pursue a healthy lifestyle [2,3]. This includes maintaining a balanced diet and undertaking some form of physical activity (PA). There is emerging evidence that PA in general might be helpful in improving motivation and the mental and physical health of transplant recipients [4].

Several studies have, however, reported that many transplant recipients do not meet the recommended amount and type of PA [5]. People are considered to be physically active when they undertake "moderate-intensity aerobic (endurance) physical activity for a minimum of 30 minutes on five days each week or vigorous-intensity aerobic physical activity for a minimum of 20 minutes on three days each week" [6]. For example, a study of 592 transplant recipients in the Netherlands found that 45% of respondents did not meet this PA threshold [7], which is substantially higher than the average level of activity of the Dutch population (28%) [8].

There is widespread recognition that many transplant recipients struggle to become and remain physically active [3,7,9–11], with evidence of reduced muscle mass [5], fatigue [11], and reduced $VO_{2peak}$ [9]. A study previously published in this journal by Van Adrichem et al. is one of the most extensive qualitative analyses of facilitators and barriers to physical exercise amongst transplant recipients (16 patients. i.e., four recipients per solid organ transplant type: heart, kidney, liver, lung), with a representative spread in terms of physical ability and exercise habits [5]. In terms of barriers, 15 out of 16 (94%) of transplant recipients experienced physical limitations in exercise, 12 (75%) reported low energy levels, and half reported comorbidities. While for some motivation was a psychological barrier to PA, for others the reinforcement, fun/pleasure and competition was a facilitator to PA.

In the present study, we compare the findings by Van Adrichem et al. [5] to a group of physically active transplant recipients, namely Transplant Athletes (TxA) who competed in cycling and triathlon events at the 24th World Transplant Games (WTG) in Perth (Australia) in April 2023. In order to perform competitively in the higher intensity cycling/triathlon events, TxA typically train for a minimum of 4–10 hours per week for at least a 12–20 week period. In addition, TxA would need to have appropriate support structures in place (such as medical, family, financial) to achieve competitive results. The main aim of this study was to

identify perceived barriers and facilitators to PA of TxA and compare these to those of Dutch transplant recipients with mixed levels of physical activity.

## Methods

### Setting and participants

In this explorative mixed methods study, we analysed data collected from 27 TxA who participated in cycling and/or sprint triathlon at the WTG 2023, and compared their lived experiences with 16 transplantees in the Netherlands in the study by Van Adrichem et al. (hereafter labelled as NL 16 transplantees) [5]. This group of 16 transplantees included a mix of both active and inactive transplantees, although their actual PA levels were not objectively assessed. Given the current study's focus on physically active transplant recipients and the research team's access to potential participants (see next section), we specifically selected cycling and sprint triathlon.In total 121 unique TxA from 27 countries were identified for inclusion who participated in cycling and/or triathlon during the WTG 2023.

Using purposeful sampling and stratification techniques, potential participants were selected based on their sex (male/female), age cohort, country, and relative performance during the cycling and/or sprint triathlon. Participants were required to have a good command of English, Dutch, or German language (languages spoken by the authors in the research team) in order to accurately respond to the interview that was conducted by one of the authors. We sought to sample TxA across a spectrum of race performances on the WTG 2023 cycling and triathlon events [12]. In the WTG 2023, cycling consisted of three separate disciplines, namely the 10 km individual time trial, 30 km individual road race, and 20 km team time trial (completed with three male athletes, or two female athletes). In total 105 cycling participants competed in at least one of these events. In the sprint triathlon a 500 meter swim was followed by a 17 km cycle and 5 km run, which could either be completed as an individual or as part of a team of three TxA. In total 44 TxA completed the individual sprint triathlon, while 15 TxA completed the team triathlon. We excluded the team time trial and the team triathlon participants who did not also compete in the cycling events, as it would be difficult to attribute individual performance. Participants were recruited in two ways. Firstly, 28 participants were approached directly in July 2023, out of which 22 (79% response rate) were subsequently interviewed. First author BR is a TxA in cycling and triathlon at WTG, and therefore has a social connection with many of these participants via social media and Online Social Fitness Networks (OSFN), notably Strava [13]. Secondly, social media platforms, including the official WTG website and several Strava and Facebook groups (e.g., Transplant Cyclists of the World, GB Transplant cyclists), were used to post an open invitation for anyone who competed in the cycling/triathlon WTG 2023 events to participate in the interview in July-August 2023. Five additional participants were recruited who were not initially sampled.

### Data collection

**Race data from world transplant games.** The results of the WTG 2023 cycling and sprint triathlon events were gathered from the official WTG website [12]. As these races were conducted with official timing chips and raced under competitive conditions (Union Cycliste Internationale-rules) on closed race circuits, we assume that these activities represent (near) the maximum athletes' physical performance capabilities. These publicly available data contain the age group and sex in which TxA competed, their finish time, average speed, and time per lap, and their absolute position (BestPos). As the data are publicly available, we will report on the absolute performance of athletes per group of 10 athletes to ensure anonymity (e.g., Top 10, Top 50) across the two disciplines.

**Online interviews.** Extensive information about the study was publicly available on a website (https://sites.google.com/view/cycling-triathlon-wtg-perth/home) and shared via e-mail to participants, and participants provided informed signed consent prior to the interview. All but one interview (Dutch) was conducted in English. All 27 transcripts were sent back to participants for sense checking, and participants could add further information if needed (five participants did).

Interviews lasted between 45 and 60 minutes (Mean = 54:44 minutes, SD = 11:04; Range 28:48–1:06:45) following the interview approach developed by [5]. During the interview 16 semi-structured questions were asked across four overlapping parts. The first part of the interview explained the purpose of the interview, providing a brief explanation of PA and high-intensity sport. The second part explored the reasons why participants joined the WTG, what their transplant journeys were (e.g., recognition of illness, transition to transplant, post-transplant), and how they were supported before, during and after transplantation by their social network, medical and social care professionals in terms of barriers and facilitators to PA. The third part of the interview focussed on participants' PA, their training intensity and purpose before and after transplant, and their satisfaction with their current level of PA. The fourth part included discussion of a visual artefact (i.e., Table 3 Van Adrichem et al [5]) of outlining the barriers and facilitators that were previously identified by Van Adrichem et al [5]. Respondents were presented with a table with barriers and facilitators to physical activity based on the Physical Activity for People with a Disability (PAD) model by [14] and extended by [5]. They were then asked–according to the categories "Personal", including physical, psychological, and other barriers/facilitators, and "Environmental" including social, physical, and other environment barriers/facilitators–whether any of the 20 barriers and 16 facilitators resonated with their own experiences, and if applicable, to link them to what they had already indicated during the interview. Participants could add any additional barriers and facilitators to the visual artefact as they saw fit. The interview concluded with an open question that allowed participants to bring up any other topic that was not discussed or clarify their previous contributions. The detailed interview procedure is described in Appendix 1.

Except for the interview with BR, which was conducted by co-author KR, interviews were conducted by BR, a WTG competitor and experienced mixed methods researcher. Interviews were conducted online using MS Teams. Audio was automatically transcribed (with explicit permission from participants). Automatic transcripts were then checked, cleaned and subsequently sent back to participants for sense checking and final agreement.

## Procedure and data analysis

This study was designed, implemented, and evaluated together with TxA in line with recommendations of Patient and Public Involvement and engagement (PPI) approach [15–18]. As argued by Holmes et al. [16] PPI is an important and expected component of health-related research activity, but there is a paucity of PPI in transplant research. By virtue of three of this study's authors (BR, PJ, NT) being TxA, this study was formulated by TxA following their own lived experiences of participating in the WTG. They noted a sense that TxA did not feel that their experiences corresponded well with the broader medical literature on transplant sport and TxA and were actively engaged in design, data collection and evaluation of this study. By combining TxA experiences and (inter)disciplinary research expertise, authors co-designed the study, implemented, and analysed the data. In line with the PPI approach, data and findings were shared and discussed with participants as part of the co-design, and findings and discussions were co-written and edited.

All available quantitative data were analysed with SPSS 27 using Mann Whitney U Tests for non-parametric testing comparing the 27 TxA with the 16 NL transplantees in Van Adrichem et al. [5]. As only aggregated data per transplant recipient group were publicly available in [5] we generated a synthetic dataset matching the exact proportions reported per respective barrier and facilitator. NVivo 12 was used for content analysis coding of interview data. BR coded all 27 interviews in their entirety to pick up on barriers and facilitators conceptualised by Van Adrichem et al. [5] and raised organically by participants in the first three parts of the interview and in the fourth part of the interview when participants were asked specifically to reflect on barriers and facilitators using the visual artefact. Subsequently, co-authors ED and LP, both unrelated to the research participants and without a role in initial data collection, independently analysed the complete transcripts of three participants using the coding approach adopted by BR. As the inter-rater reliability with one coder was initially relatively low, a follow-up online meeting was arranged with the three coders to clarify the procedure of coding. The final inter-rater reliability between the three coders ranged between 0.72–0.83 (Cohen's Kappa), indicating substantial agreement between the coders.

This research received Human Ethics Research Approval (HREC/4787/Rienties). Participants were free to participate and withdraw their consent at any time. No consent was withdrawn. The COREQ checklist [19] is available in Online Appendix 2.

## Results

### Study population

Table 1 lists all 27 participants and their type of transplant and when they received their first transplant. In order to ensure anonymisation we classified participants based on continent of origin and rounded the best position in their respective races based on a factor of 10 (e.g., P6 was in the Top10 of best performing TxA, while P20 was in the Top30). Participants were from nine countries across four continents (18 Europe, 6 Australasia, 2 North America, 1 Africa). Of the 27 participants who were interviewed, four (14%) identified as female TxA and 23 (86%) identified as male TxA. 96% participated in cycling, 37% participated in sprint triathlon, and 33% participated in both disciplines.

In order to check for potential non-response bias, we compared age, sex, and training intensity as measured by publicly available Strava data of the 27 participants with the 94 TxA who did not volunteer for an interview, but no statistically significant differences were found. However, using a Mann-Whitney U Test those who participated in the interview on average had a better finishing position in the cycling and/or triathlon events (z = -3.823, p < .001, r = .34), with a medium effect size relative to those who did not participate in this study. While participants in this study had comparable demographic characteristics and similar recorded training activities as other TxA, perhaps it is not surprising that relatively higher performing participants participated in the study as they were keen to share their "success stories" [20]. Nonetheless, our sample includes a range of performances similar to the broader cycling and triathlon TxA population. Therefore, we argue that a representative mix of diverse high intensity TxA participants was present in this study.

### Identification of barriers and facilitators

When comparing the participants in the current study to the 16 NL transplantees in Van Adrichem et al. [5], our participants were slightly older (54.4 years vs. 50.5 years) and more likely to identify as male (85% vs. 56%). The 16 NL transplantees included four recipients for every type of solid organ transplant (i.e., heart, lung, liver, kidney), while our TxA study included a

**Table 1. Basic descriptors of 27 transplantees (based upon type of transplant and year of transplant).**

| Code | Sex | Age Group | Continent of Birth | Type of Transplant | Sport | Best Position | When transplanted |
|------|-----|-----------|--------------------|--------------------|-------|---------------|-------------------|
| P6 | Male | 60–69 | Europe | Bone marrow | C + S | Top 10 | 2005 |
| P16 | Male | 30–39 | Europe | Bone marrow | C | Top 20 | 2010 |
| P17 | Male | 30–39 | Europe | Bone marrow | C + S | Top 10 | 2010 |
| P2 | Male | 60–69 | Europe | Bone marrow | C | Top 20 | 2012 |
| P27 | Female | 30–39 | Australasia | Bone marrow | C | Top 80 | 2012 |
| P3 | Male | 40–49 | Australasia | Bone marrow | C | Top 10 | 2013 |
| P22 | Male | 50–59 | Europe | Bone marrow | C + S | Top 40 | 2017 |
| P14 | Male | 60–69 | Europe | Bone marrow | C + S | Top 40 | 2020 |
| P9 | Male | 40–49 | Europe | Bone marrow | C | Top 10 | 2021 |
| P19 | Male | 50–59 | Europe | Heart | C + S | Top 10 | 2012 |
| P20 | Male | 60–69 | Europe | Heart | C | Top 30 | 2014 |
| P12 | Male | 18–29 | Australasia | Heart | C | Top 30 | 2021 |
| P11 | Male | 50–59 | Europe | Kidney | C | Top 70 | 1999 |
| P5 | Female | 18–29 | Australasia | Kidney | C | Top 70 | 2000 |
| P10 | Male | 50–59 | North America | Kidney | C | Top 40 | 2000 |
| P4 | Male | 50–59 | North America | Kidney | C + S | Top 10 | 2005 |
| P1 | Male | 40–49 | Europe | Kidney | C + S | Top 10 | 2008 |
| P13 | Female | 40–49 | Europe | Kidney | S | Top 30 | 2009 |
| P23 | Male | 50–59 | Europe | Kidney | C | Top 20 | 2014 |
| P24 | Male | 60–69 | Europe | Kidney | C | Top 10 | 2015 |
| P15 | Male | 30–39 | Europe | Kidney | C | Top 20 | 2017 |
| P18 | Female | 50–59 | Australasia | Liver | C | Top 50 | 1999 |
| P8 | Female | 50–59 | Europe | Liver | C | Top 60 | 2002 |
| P25 | Male | 40–49 | Australasia | Liver | C + S | Top 10 | 2013 |
| P7 | Male | 18–29 | Europe | Liver | C | Top 20 | 2015 |
| P21 | Male | 50–59 | Africa | Liver | C + S | Top 20 | 2018 |
| P26 | Male | 50–59 | Europe | Liver | C + S | Top 10 | 2021 |

Note: n = 27. Numbers in the scatterplot refer to respective participant. Scale from 0% (no barriers, no facilitators) to 100% (all the barriers, all the facilitators) as indicated in Table 2.

wider range of transplants with three heart, six liver, nine kidney, and nine bone marrow transplant recipients.

Table 2 points to several substantial differences in reported barriers and facilitators between the 16 NL transplantees and the 27 TxA in our study. Many of the barriers reported by the 16 NL transplantees were not shared by most of the TxA, while more facilitators were reported by the TxA. As detailed below, the most important differences were significantly lower physical limitations, lower fear of exercise, and no comorbidity for TxA relative to the 16 NL transplantees. Furthermore, the 27 TxA perceived to have substantial social support and muscular strength to do PA, and were in control of their weight, although several reported a lack of clarity around the level of support and understanding from their medical professionals.

Subsequent analyses per TxA did show some differences in terms of lived experiences, as indicated in Fig 1. Many TxA were positioned on the bottom right of Fig 1 with relatively few or no experienced barriers, and primarily experienced mostly facilitators to PA. For example, P1, P7, P17, P19, or P25 experienced no personal barriers whatsoever, although as will become evident in the in-depth qualitative analysis below some TxA did talk about "invisible handicaps" (see section *Physical limitations*) and limited medical support for high-intensity sport

**Table 2. Comparison of reported barriers and facilitators of PA for 27 TxA vs 16 NL transplantees (in percentages).**

| | Barriers | | | | | | | | | Facilitators | | | | | |
|---|---|---|---|---|---|---|---|---|---|---|---|---|---|---|---|
| | **1** | **TxA** | **NL16** | **2** | **TxA** | **NL16** | **3** | **TxA** | **NL16** | **4** | **TxA** | **NL16** | **5** | **TxA** | **NL16** |
| Stronger | Physical limitations | 26** | 94 | Side-effects of medication | 30 | 38 | Group activity | 96** | 25 | Routine/habit | 100 | 88 | Motivation | 100 | 100 |
| | Fear | 26 | 56 | Energy levels | 26** | 75 | Self-efficacy | 89 | 81 | Social support | 96** | 56 | Consequences of (in)activity | 44** | 88 |
| | Bad weather | 22 | 38 | Social role | 7 | 31 | Expertise of personnel | 74 | 81 | Coping | 93 | 94 | | | |
| | Age | 22 | 19 | | | | | | | Strength | 89** | 50 | | | |
| | Post-transplant life events | 19 | 25 | | | | | | | Goals/goal priority | 85 | 81 | | | |
| | Financial resources | 15 | 19 | | | | | | | Weight | 67 | 31 | | | |
| Weaker | Comorbidity | 0** | 50 | | | | | | | Transplanted organ | 33 | 63 | | | |

Note: TxA = 27, n NL 16 = 16. Mann Whitney U test, * p < .05, ** p < .01.

(see section *Medical and professional support for PA*). At the same time, there were several TxA who did report numerous barriers (e.g., P5, P8, P10, P11, P22, P27). Therefore, in the next section we will follow the structure of Table 2 and discuss each category in turn to explore the lived experiences of the 27 TxA as well as to compare these with the 16 NL transplantees.

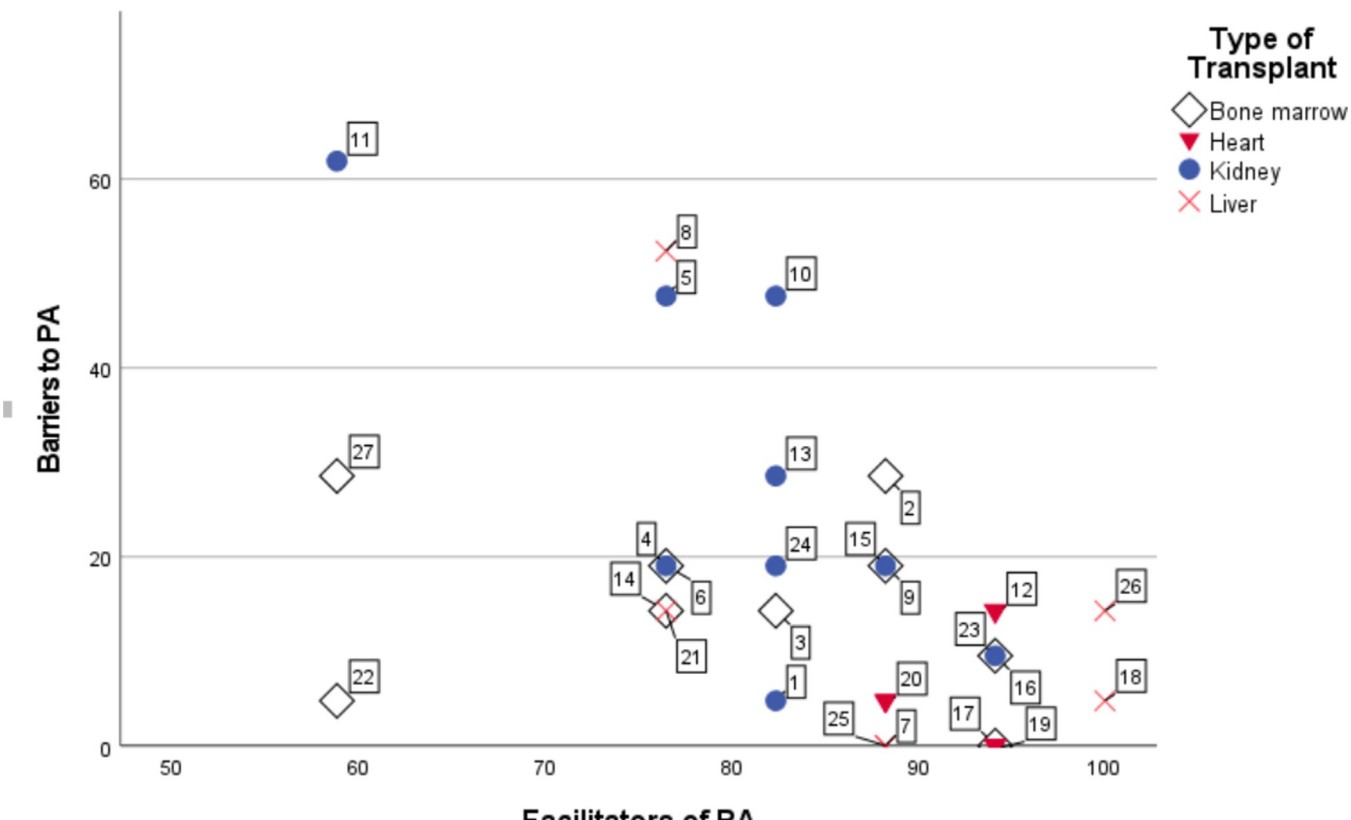

**Fig 1. Reported barriers and facilitators of transplant athletes to do physical exercise (%).** Note: n = 27. Numbers in the scatterplot refer to respective participant. Scale from 0% (no barriers out of possible 20; no facilitators out of 16) to 100% (20 out of 20 barriers; 16 out of facilitators).

## Barriers to physical activity

**Physical limitations.** As indicated in Table 2, 7 out of 27 (26%) high intensity TxA reported physical limitations to PA, while 15 out of the 16 (94%) NL transplantees reported physical limitations, which is a statistically significant difference ($z = -4.027$, $p < .001$, $r = .61$) whereby the effect size r indicated a large effect. During the interviews, TxA self-reported to exercise on average 8.33 hours per week (SD = 3.09, range 1–11), which is substantially more than an average adult [8,20]. In another study [21], we demonstrate that the TxA self-reported information about training load is highly correlated with their training data from Strava.

Despite comparatively high average PA, several TxA did report some physical limitations after transplant, and also expressed these (perceived/real) physical limitations in slightly different ways. For example, P5 indicated that she carefully selected cycling races that she thought she could complete given her physical limitations. P5 is an experienced cyclist who raced at an amateur level during her first transplant, but she reported that her second transplant had resulted in a more difficult recovery and increased barriers to PA, which was also reflected in her relatively lower racing performance (i.e., Top 70).

*"Physical limitations because they are constantly on my mind like I am assessing "Is this a race I can't complete, you know. For example, a 70 kilometer road race with no laps" And like I'm probably not going to be able to complete that. Ohh, and lack of energy. It's whenever I struggle. It's always just a sort of hitting a wall with energy." (P5, female, kidney, 30–40, 31:08, Top 70)*

Another participant (P17) indicated that in a way transplant people have an "invisible handicap". P17 is one of the Top 10 TxA who is an active participant in sport, but with the amount of training he does he still felt that he was not able to compete at an equal level with able-bodied athletes:

*"Sometimes I feel like I'm not running or cycling at the best of what I could do, given especially the volume of training I have, but even with other thing and especially with COVID, it really highlighted the fact that we are immunosuppressed in some capacity, and we have to be more careful. People they don't really realise how sensitive of a population we are and that sort of like invisible handicap . . . (P17, male, bone-marrow, 30–40, 38:38, Top 10).*

**Lack of energy.** Similarly, while 75% of the 16 NL transplantees reported a lack of energy after transplant, only 26% of TxA reported the same barrier ($z = -2.849$, $p < .001$, $r = .43$), which again is a statistically significant difference with a medium effect size. Most TxA indicated they have sufficient energy to combine a busy life with extensive PA. Nonetheless, some participants did indicate a lack of energy. For example, P8 indicated that she needed to carefully balance her energy in her household, and regularly took "nanna naps".

*"Lack of energy. Yeah, I've started having my Nanna Naps in the afternoon. But that's all. Since the transplant I'm in bed by nine o'clock most nights. I'm absolutely shattered." (P8, female, liver, 50–60, 31:07, Top 60)*

P27 was one of the few TxA who reported that even 10 years after her bone marrow transplant she still experienced a lack of energy, and often had to take naps in the afternoon. While her employer was fully supportive and provided flexible working hours, P27 nevertheless experienced several physical limitations and a lack of energy.

*"I just seem to crash in the afternoons. It does really matter what I do. So I can get through my work day fine. But you know, depending on what's going on in my life, I nap a lot to kind of get some of that energy back"* (P27, female, bone marrow, 30–40, 28:11, Top 80)

**Comorbidity.** Rather surprisingly none of the TxA in our study experienced comorbidity issues, even after prompting by the interviewer, while half of the 16 NL transplantees reported comorbidity issues ($z = -4.025$, $p < .001$, $r = .61$), with a large effect size. In fact, several (P12, P21, P22, P26) reported that they were able to reduce or completely withdraw some or all their (e.g., high blood pressure, beta blockers) medication when increasing their PA levels. Several bone-marrow TxA were able to cease immunosuppressant medication, while for the kidney, heart, and liver TxA this was not feasible.

*"I think exercise plays a major role in terms of managing the comorbidities. I mean before my transplant, I was an insulin dependent diabetic. I've managed to get off the insulin, so I'm now only on oral medication.". (P22, male, liver, 50–60, 42:05, Top 20)*

*"Beta blockers, now that's the one you don't want to take, so you know it was one of the main goals at the beginning for me to get active as soon as possible, trying to get rid of those medications. And after three months, I could leave the beta block. And after six months, I could leave the other one and everything was relatively going back to normal". (P26, male, liver, 50–60, 09:09, Top 10)*

**Fear of exercise.** 26% of TxA reported fears of exercise, injuries or falling off the bike, while 56% of the 16 NL transplantees reported concerns ($z = -1.706$, $p > .05$). Most TxA participants reported that they felt very comfortable and safe riding their bike, running, or swimming, and did not worry about crashes, injuries, or unsafe sporting conditions. Some, like P23, even considered it as part of the joy of the cycling experience.

*"No, love it. No, I fall off quite regular. . . I've not broken anything yet, but yeah, I come off quite often, yeah. . . I just bounced. . . I think the main times where you're training hard, you're pushing maybe a little bit too hard in the conditions (P23, male, kidney, 50–60, 12:29, Top 20).*

Others did express some fear of riding their bike, and/or exercising intensively. For example, P27, a comparatively less experienced TxA cyclist, reported anxiety with high heart rates when riding her bike during WTG, and being afraid to get into a bike accident due to an unrelated sporting incident.

*"And this was a bit of a concern for me that when I was cycling on that first day [of WTG], my heart rate got really high. I don't know what levels I should be out or what's, you know, my limit. I just know I felt that it was too high for what I normally do when I'm exercising." (P27, female, bone marrow, 30–40, 16:08, Top 80).*

**Other barriers (bad weather, post-transplant events, age, financial resources).** In terms of the other barriers reported by Van Adrichem et al. [5], no significant differences were found that related to bad weather, post-transplant life events, age, or financial resources. In other words, other barriers were of similar importance to the 27 TxA as they were for the 16 NL

transplantees. For example, cycling outside in the rain was broadly perceived not to be a pleasant experience and around half of TxA [14] reported this to be a barrier (noting they would ride indoors on Zwift, an online training platform, during bad weather). This proportion was not statistically different as reported in Van Adrichem et al. [5].

## Barriers that could be facilitators (and vice versa)

Following Van Adrichem et al [5] three factors (i.e., self-efficacy, medical and professional support, and riding in groups) could serve as barriers for some but facilitators for others.

**Self-efficacy and medical and professional support for PA.** There was no significant difference in self-efficacy constituting a barrier or facilitator between the 27 TxA and 16 NL transplantees. In terms of self-efficacy most TxA were comfortable with PA and in particular riding their bike or doing sprint triathlon activities.

Equally, there was no significant difference in the experience of medical and professional support for PA between the 27 TxA in this study and 16 NL transplantees. 18 TxA indicated that their medical and social care professionals were helpful (in some way) in supporting their transition to PA and becoming a high-intensity TxA. Nonetheless, several participants indicated that most PA advice was rather generic and geared towards recreational activity, and not to high-intensity TxA.

*"In the end, no one said I should do Ironman with that, so definitely. And I mean, you get this brochure, in my case "What to eat?" And what do you have in your living room? And then there's like good sport activities are swimming, cycling and jogging. Yeah. So I just did that and no one said if I should do like 180 kilometres [of cycling] and run a marathon after that. . . I think they, in the beginning, some doctors said "Ohh. No, that's not good". (P19, male, heart, 50–60, 11:28, Top 10).*

**Riding in groups.** All except one TxA (96%) indicated that they had plenty of opportunity to exercise with other people in a group, whereas just 25% of the 16 NL transplantees reported the same level of opportunity (z = 4.199, p < .001, r = .64), indicating a large effect size. Some TxA indicated they formed cycling groups with their local friends (P21), while others joined a local cycling club (P15) or rode together with other TxA at a regional/national level (e.g., P8). For example, P21 indicated that all of his friends were directly involved in cycling, and that he would ride on the weekends with them, and go on cycling holidays together.

*"Normally I don't have so much time during the week, but normally at the weekends and Sundays I ride with my friends here. My friends are my friends from cycling, so they support to do that" (P21, male, bone-marrow, 50–60, 38:27, Top 40).*

P15 initially rode with other TxA but also joined his local cycling club.

*"At the beginning when I started cycling I was like I want to compete with the other transplanted people. I'm most of the time the road captain, so I'm sitting on the front of the bunch with my race [armband].Most of the time, if there are new people, they think I'm a doctor or something of a medical guy because "Your legs are too good!" My [local cycling club], they know I'm transplanted, but most of the time they know me as "OK. Watch out for P15, he gonna do the sprint!". (P15, male, kidney, 30–40, 23:55, Top 20).*

## Facilitators

In terms of facilitators, TxA participants were generally much more positive about facilitators of PA than the 16 NL transplantees. For many TxA, PA was described as a routine/habit, a high level of (perceived) social support for their PA, a coping mechanism, a goal, a way to gain strength, and means to maintain weight.

**Routine/Habit.** All TxA reported PA to be a routine or habit, compared to 88% of 16 NL transplantees, that helped to structure their life. For example, for P17 doing exercise gave him a sense of purpose and achievement.

*"I've always seen in my entire life sport being just an enabler, so I don't need motivation. I'm just self-motivated in that sense. I'm just excited to go out and become a better person. I always will make time for it. It's my priority." (P17, male, bone-marrow, 30–40, 28:59, Top 10)*

**Social support.** Nearly all TxA (96%) indicated high levels of social support from family and friends to participate in PA ($z = 3.224$, $p < .01$, $r = .49$) which was significantly higher than the 16 NL transplantees (56%) with a medium effect size. For example, P25 indicated that both before, during, and after his transplant he received a lot of support from his social environment, and to continue with PA.

*"I have got a great amount of support from my immediate family, of my wife and kids, but broader than that, my parents, my in-laws, my brothers and sisters, so I had a lot of support there."(P25, male, liver, 40–50, 6:37, Top 10)*

At the same time, P24 who used to be a semi-professional cyclist in his younger years reflected that now with family obligations at times it is more difficult to just take time to do long intensive PA.

*"I do think to be a good sportsman you have to be quite selfish, and it's a lot harder now for me as a you know as somebody who's married with a wife and also has got two sons with their partners and, you know, all the other family connections, there's a lot harder for me to dedicate myself to training then it was when I was 18 and living at home. I have joined [local cycle club XXX] after my transplant, and they're a very good club and I get a lot of social support from them" (P24, male, kidney, 50–60, 24:55, Top 10)*

P13 indicated that many of her friends before transplant could not really relate to her new active post-transplant life and she found more social support from other TxA.

*"I have made a lot of really close friends through transplantation and sport because everyone has been through similar experiences and they can really relate to you, whereas friends from the past don't have the same concept of understanding of what a transplant patient and athlete goes through." (P13, female, kidney, 40–50, 19:13, Top 30)*

**Coping.** For most TxA (93%), as for the majority of the 16 NL transplantees (94%) PA helped them in coping with their busy lives. Several participants mentioned that riding their bike, going for a run, or a swim, allowed them to clear their head and find a way to release their stress. For example, P12 indicated that cycling allowed him to cope with the daily pressures of having a busy professional life as well as a young family.

*"But the motivating factors of coping as a big one. I find cycling really good for just being in your own mind and clearing your head a bit." (P12, male, heart, 30–40, 38:07, Top 30)*

P22 used PA to cope with the busy professional and family life.

*"The facilitators I think it, especially with the coping. I mean if I'm having a bad day, an hour on the bike will really help me feel better. I even tell my friend, let me just go bash myself on the bike and I'll feel better after that." (P22, male, liver, 50–60, 44:51, Top 20)*

**Strength.** Most of the TxA (89%) indicated that gaining strength was an important driver for PA, while only half of the 16 NL transplantees (50%) indicated this (z = 2.792, p < .01, r = .43), which again is a statistically significant difference of medium effect size. For example, P26 described their drive to become stronger.

*"It's always a goal to get better, to get stronger. It's for me mostly the muscle strength." (P26, male, liver, 50–60, 27:47, Top 10)*

**Goals/Goal priority.** For a large group of TxA (85%) having clear goals was essential for doing PA, which was similar to the group of 16 NL transplantees (81%). Those who had relatively high training loads of > 6 hours per week reported goal-setting to be especially important. Knowing that a large event like an Ironman or a WTG was coming up was a motivator for several TxA. For P18, having clear goals for an event like the WTG was important.

*"I find from a mental wellbeing perspective, I have to be active, whatever that means. So yeah, I'm not good if I don't. I do find having a transplant games or competition goal really important cause I ebb and flow. I definitely increase my intensity and commitment when I've got that goal. But I try to set smaller goals that aren't quite so distant because sometimes it's a year out." (P18, female, liver, 50–60, 46:45, Top 50)*

**Weight.** Two thirds of TxA indicated that they did not have any issues with weight. The nature of endurance sports means that athletes have substantial opportunities to maintain a healthy weight. In fact, for several TxA, the PA allowed them to eat a bit more and enjoy life, as they knew they would be able to burn any additional calories through their PA.

*"Not the weight. The weight is OK and I was feeling better than when I exercise. It's increased my energy levels, of course, yes. But with the sports, of course you maintain the weight and you can eat a little more." (P21, male, bone-marrow, 50–60, 51:30, Top 40)*

**Transplanted organ.** Only one third of TxA (33%) indicated that they exercised, in part, to honour their transplanted organ. By comparison, 63% of the NL transplantees indicated that this resonated with them, though this difference was not significant (z = -1.840, p > .05). This is not to say that two thirds of TxA did not honour their donor and/or their gift of life. However, for them, the motivation for participating in PA and competing in the WTG was not necessarily due to their transplanted organ. Most TxA did not talk about their donor unprompted during their interview, and when prompted by the visual artefact they often did

not indicate that their transplanted organ was an important driver for PA. For example, P13 indicated:

*"I wouldn't say that my donor is a motivator for me and that sounds awful. But it's not. It's not that I'm not grateful. I'm immensely grateful. And there's always a tear, and I do get emotional at the games when, you know the donor families [are] coming out. I do. But I don't stand on the start line and think of my donor, if I'm being honest." (P13, female, kidney, 40–50, 12:18, Top 30)*

**Motivation.**   As expressed by all the TxA there was often a strong intrinsic motivation to exercise. This is the same for the 16 NL transplantees. TxA indicating that riding was also for fun and pleasure.

*"Riding your bike is primarily fun. Competitiveness is a bonus. And the more you do, the more you can do. That's the more you do, the easier it is to do the things you want to do. (P20, male, heart, 60–70, 43:48, Top 30).*

Others like P3 rode for the feeling of exhilaration of pushing oneself to the absolute limit after recovering from illness.

*"You go through the grief cycle of coping with the diagnosis and the treatment. And one of those elements of that process is bargaining, isn't it? And then I remember thinking, you know, I would give anything to feel that pain in my legs and the burning in my lungs, of riding in a race or up a steep hill, pushing myself 100%. And I promised myself that if I ever got the privilege to be able to do that again, I would enjoy it and I would appreciate it. Whenever I'm at that absolute limit I feel this surge of kind of emotional energy come through me, that's such a privilege to be able to feel that and to choose to feel that I can push harder. (P3, male, bone marrow, 40–50, 15:59, Top 10)*

**Consequences of (in)activity.**   Relative to the 16 NL transplantees the TxA participants were less worried about the consequences of inactivity (z = 2.759, p < .01, r = .42). For example, as indicated by P19:

*"I mean the most, the most important thing I think that the way I do it, it gives me more years to live because it's like a lifestyle" (P19, male, heart, 50–60, 51:49, Top 10)*

Therefore, many TxA were less inclined to live a sedentary or perhaps unhealthy lifestyle. As mentioned by one TxA who used to be sedentary before his transplant:

*"I think the question of habit, I mean it becomes part of who you are or what you do, so that and then also the consequences of inactivity. I mean I don't want to be inactive. I've seen what it does." (P22, male, liver, 50–60, 45:55, Top 20)*

**Joy of competition.**   Finally, an important theme for TxA was the joy of preparing for and competing in events at a local, national and/or international level, including the WTG. Competition was an important driver for several to push themselves more during training.

*"I guess since I started to participate in the [blinded] transplant games, seeing the level of competition. And of course, when I started to win medals, initially bronze and then eventually silver and then gold, I realised that I had to be quite fit in order to compete. I probably train around 10 hours a week and following a training programme. I'm definitely training more because I want to be at a good level when I'm competing at these sports. When I was cycling before I got diagnosed, I was just a good but average cyclist, not necessarily competing for world medals."* (P1, male, kidney, 40–50, 39:06, Top 10)

Like P1, others (P23) gained a lot of motivation thinking about their next competition event, and thinking about their competitors and whether they should "be sitting on their sofa" rather than adding another training session.

*"[We] talked about motivational factors of what you actually training [for]. I kind of hit on that little bit before, but it's not a joke. It is very, very serious. One, motivation, I know [exercise] distresses me, so that's really good. But the other motivations I know my competitors and think about people like [P1], [P9], and [P19] . . . they won't be sitting on the sofa. So there is motivational factors that some of the competition and the people that you've compete with and against get me off the sofa and get me either in the pool or get me on the bike.* (P23, male, kidney, 50–60, 25:17)

## Discussion

This explorative mixed methods study compared the lived experiences of 27 physically active Transplant Athletes (TxA) who participated in cycling and/or sprint triathlon at the WTG 2023 with a group of 16 NL transplantees [5] who represent a broader spectrum of PA amongst transplant recipients. There is now a substantial body of work [1,5,7,9,10] that indicates that many transplant recipients do not meet the recommended amount and type of PA. Furthermore, a range of studies [2–5,7,9–11] found several important barriers and facilitators to PA, including physical limitations, lack of energy, and lack of medical and social support that limited transplant recipients perceived ability to become and remain physically active.

Our main aim was to identify the perceived barriers and facilitators to PA of TxA and compare these to those of transplant recipients with average levels of physical activity. The most important findings of this study were significantly fewer barriers to PA for TxA relative to the NL 16 transplantees, in particular fewer physical limitations, lower fear of exercise, and lack of comorbidity. While all but one of the NL 16 transplantees indicated that they experienced physical limitations to exercise, 74% of TxA reported no physical limitations. Another barrier mentioned by 75% of the NL 16 transplantees was a lack of energy to perform PA, while only 26% of TxA reported this same barrier. Lack of energy was a barrier for a minority of TxA (26%) with most indicating they have sufficient energy to combine a busy life with extensive PA.

From a medical perspective, it was surprising that none of the TxA reported experiencing any comorbidity issues, while half of the NL transplantees indicated this restricted their PA. Particularly notable are experiences of being able to reduce or completely withdraw some or all their medication when increasing their PA levels (e.g., blood pressure medications, beta blockers).

Overall TxA participants were much more positive about the facilitators of PA than the NL 16 transplantees. Several participants mentioned that PA allowed them to clear their head and find a way to release stress. PA was seen as a routine and a habit that helped to structure their life, while for others their lives were more structured around PA. Due to the intensity of PA,

most TxA did not have any weight issues which contrasts with the wider transplant community [5,7] and adult population in general [8]. Furthermore, TxA reported more social support for their PA relative to the NL 16 transplantees.

Relative to the NL16 transplantees, TxA's motivation to do PA was less driven by their transplanted organ and consequences of (in)activity, which might be a result of the fact that most of TxA had internalised PA as part of their healthy lifestyle. As expressed by many of the TxA, there was often a strong intrinsic motivation to exercise, as participants indicated that exercising was something they also did for fun and pleasure. Several TxA also indicated that pushing themselves to the absolute limit after recovering from illness was a great feeling. In line with previous findings [9,11], competition was an important driver for several TxA to push themselves more during training, while for others, the ability to compete was an indirect bonus of being physically active. While competition was an important driver to train and participate in the WTG for some, for others it was the opportunity to meet like-minded people and to travel to enjoy the gift of life. Whilst TxA generally reported that their medical professionals/care team were supportive, there appears to be a lack of specific medical advice or encouragement to participate in high intensity sports.

Most TxA reported to exercise eight or more hours per week, mostly participating in intensive sports such as cycling, running and/or swimming, which is substantially more than typical able-bodied adults [2,6,8]. Some transplant athletes were able to perform PA at higher levels of intensity than traditionally recommended by medical professionals. While there are no formal or objective assessments of fitness at any stage post-transplant in most medical centres across the globe, our study suggests that for some transplantees active encouragement to exercise can have both positive physical and medical effects. Obviously, there may be a self-selection effect present in those who are able to compete in high intensity sports at WTG like cycling or triathlon due to a more positive medical, social, and PA journey compared to other transplantees. Furthermore, these findings might not be generalisable across a wider group of TxA, or transplant recipients in general. Indeed, [20] warn that the potential affordances and abilities of transplant athletes to compete with high levels of PA might not necessarily translate to a wider and more diverse group of transplant recipients. Furthermore, with a lack of longitudinal studies on the long-term impact of high-intensity levels of PA on health [20] "emphasise that while exercise can serve as a potent therapeutic intervention after transplantation and is probably underutilized in the transplant population, the line between its medicinal benefits and potential harm lies in the dosage administered".

Nonetheless, our study does provide important insights for other transplantees and the medical and social care profession as it shows that with determination, time, and appropriate support some TxA are able to cycle, run and/or swim at a high level of PA intensity. In fact, several interviewed TxA who were sedentary before transplantation, have completed Ironmans and/or performed competitively with able-bodied athletes in local and national events. Rather than just focussing on surviving a life-threatening illness, our study may support medical professionals and transplantees alike to realise that TxA can often do much more than just basic walking or perhaps an occasional run.

An obvious limitation of this study is the relatively small sample size of 27 TxA. Furthermore, those who were able to compete at a world level in cycling and triathlon might not necessarily be representative for the wider transplant community. With a larger sample size future research might be able to identify more distinct patterns in PA and perceived barriers and facilitators. Finally, all interview data were obtained and coded by the BR, thereby potentially introducing researcher positionality [17]. However, in line with PPI we extensively involved TxA participants in this study, and used not only self-reported interview data but also triangulated these with objective race data. Furthermore, two authors completely independently from

the data collection process independently analysed three transcripts, and confirmed inter-rater reliability of the coded barriers and facilitators to PA.

## Conclusion and recommendations for future research

The findings from this study demonstrate that there is significant variation in the barriers and facilitators to PA between transplant recipients who compete in high intensity sports such as cycling and sprint triathlon, and the broader transplant recipient population, often with a medium to large effect size. Most of the TxA in this study reported fewer barriers related to physical limitations, fear of exercise and comorbidity relative to previous studies of transplant recipients. TxA also reported a broader range of facilitators that often related to the pleasure, and positive psychological effect that they derive from participating and competing in their chosen high intensity sport. TxA also reported high levels of support from their social networks, family.

While these insights provide a powerful narrative that some groups of TxA are able to become and remain very physically active, we are not expecting or anticipating that all transplantees would be able to reach similar fitness levels if they trained 8+ hours per week, and whether this would be advisable [20]. Future research could be undertaken to further explore the supporting factors and limitations to athletic performance in TxA in order to establish evidence-based guidelines of what transplantees with appropriate support can achieve. Furthermore, it would be useful in future research to determine why some TxA develop strong identities around their transplant sport and transplant identity, and how motivation and the social environment around TxA might support or hamper their journeys.

## Supporting information

**S1 File. Supplementary file consolidated criteria for reporting qualitative studies (COREQ).**
(DOCX)

**S2 File. Supplementary file data of barriers and facilitators of physical activity.**
(XLS)

**S3 File. Supplementary file indicative semi-structure interview questions.**
(DOCX)

## Author Contributions

**Conceptualization:** Bart Rienties, Ben Oakley, Keetie Roelen, Nicholas Topley.

**Data curation:** Bart Rienties, Keetie Roelen.

**Formal analysis:** Bart Rienties, Elaine Duncan, Liset H. M. Pengel, Keetie Roelen.

**Investigation:** Bart Rienties, Perry Judd, Keetie Roelen.

**Project administration:** Bart Rienties.

**Resources:** Bart Rienties.

**Supervision:** Ben Oakley.

**Writing – original draft:** Bart Rienties, Ben Oakley, Keetie Roelen.

**Writing – review & editing:** Bart Rienties, Elaine Duncan, Perry Judd, Ben Oakley, Liset H. M. Pengel, Keetie Roelen, Nicholas Topley.

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
