## [Decision Letter · Decision Letter 0]

8 May 2024

PONE-D-23-39099Barriers and facilitators to physical activity: a comparative analysis of transplant athletes competing in high intensity sporting events with other transplant recipients.PLOS ONE

Dear Dr. Duncan,

Thank you for submitting your manuscript to PLOS ONE. After careful consideration, we feel that it has merit but does not fully meet PLOS ONE’s publication criteria as it currently stands. Therefore, we invite you to submit a revised version of the manuscript that addresses the points raised during the review process.

We look forward to receiving your revised manuscript.

Kind regards,

Jianhong Zhou

Staff Editor

PLOS ONE

3. Please remove your figures from within your manuscript file, leaving only the individual TIFF/EPS image files, uploaded separately. These will be automatically included in the reviewers’ PDF.

Reviewers' comments:

Reviewer's Responses to Questions

**Comments to the Author**

1. Is the manuscript technically sound, and do the data support the conclusions?

Reviewer #1: Yes

Reviewer #2: Yes

2. Has the statistical analysis been performed appropriately and rigorously? 

Reviewer #1: Yes

Reviewer #2: Yes

3. Have the authors made all data underlying the findings in their manuscript fully available?

Reviewer #1: Yes

Reviewer #2: Yes

4. Is the manuscript presented in an intelligible fashion and written in standard English?

Reviewer #1: Yes

Reviewer #2: Yes

5. Review Comments to the Author

Reviewer #1: Your manuscript presents an insightful mixed-methods study comparing transplant athletes with average transplant recipients regarding physical activity barriers and facilitators. This work contributes to the understanding of physical activity in transplant recipients, a vital area of research. However, I have identified several areas that could be strengthened to enhance the clarity, rigor, and impact of your study.

Statistical Analysis and Reporting:

while the statistical methods are standard and the coding process for qualitative data appears robust, the manuscript could be strengthened by providing more detail on the checking of statistical assumptions, explaining the rationale behind the sample size, and including effect sizes and confidence intervals. Additionally, normality and homogeneity of variances.

It's not clear whether a power analysis was conducted to determine the sample size for quantitative comparisons, which raises questions about the study's power to detect meaningful differences.

Generalizability of Findings:

Consider discussing the limitations regarding the generalizability of your findings. While your study provides valuable insights, the specific nature of your sample may limit its applicability to other transplant recipient populations.

Reviewer #2: Review comments on manuscript entitled "Barriers and facilitators to physical activity: a comparative analysis of transplant athletes competing in high intensity sporting events with other transplant recipients"

Comments to the Author

Thank you for the opportunity to review this study. The idea of research is very interesting, organized and well written reasonable. The authors have done great effort to accomplish this work. However, there are some comments and suggestions.

Section-by-section comments

Title: Well structured, concise and informative.

Abstract:

- It is recommended to write the abstract in sections including (background, aim, methods, results, conclusion)

- It is recommended to write 5-7 keywords in alphabetical order.

Introduction :

- Well structured

Methodology:

- Well structured

Statistical analysis:

- Well structured

Discussion:

- Well structured

6. PLOS authors have the option to publish the peer review history of their article (what does this mean?). If published, this will include your full peer review and any attached files.

Reviewer #1: No

Reviewer #2: No

---

## [Author Response · Author response to Decision Letter 0]

22 May 2024

Please find attached our revision document.

---

## [Decision Letter · Decision Letter 1]

10 Jun 2024

PONE-D-23-39099R1Barriers and facilitators to physical activity: a comparative analysis of transplant athletes competing in high intensity sporting events with other transplant recipients.PLOS ONE

Dear Dr. Duncan,

Thank you for submitting your manuscript to PLOS ONE. After careful consideration, we feel that it has merit but does not fully meet PLOS ONE’s publication criteria as it currently stands. Therefore, we invite you to submit a revised version of the manuscript that addresses the points raised during the review process.

 Please note that I served as the 2nd reviewer on the resubmission, and as the 2nd academic editor I did not provide feedback on the first submission. In my comments below I noted several areas where the manuscript does not currently meet the publication criteria (e.g., COREQ checklist, data availability, detail required on statistical analysis). 

We look forward to receiving your revised manuscript.

Kind regards,

Heather Macdonald, Ph.D

Academic Editor

PLOS ONE

Additional Editor Comments:

Please note that the first academic editor on this manuscript was not available to handle the revision, so it was transferred to me. I also served as the 2nd reviewer as one of the reviewers on the first submission was not available to review the resubmission.

Reviewers' comments:

Reviewer's Responses to Questions

**Comments to the Author**

1. If the authors have adequately addressed your comments raised in a previous round of review and you feel that this manuscript is now acceptable for publication, you may indicate that here to bypass the “Comments to the Author” section, enter your conflict of interest statement in the “Confidential to Editor” section, and submit your "Accept" recommendation.

Reviewer #2: All comments have been addressed

Reviewer #3: (No Response)

2. Is the manuscript technically sound, and do the data support the conclusions?

Reviewer #2: Yes

Reviewer #3: Partly

3. Has the statistical analysis been performed appropriately and rigorously? 

Reviewer #2: Yes

Reviewer #3: No

4. Have the authors made all data underlying the findings in their manuscript fully available?

Reviewer #2: Yes

Reviewer #3: No

5. Is the manuscript presented in an intelligible fashion and written in standard English?

Reviewer #2: Yes

Reviewer #3: Yes

6. Review Comments to the Author

Reviewer #2: Review comments on Manuscript Number: (PONE-D-23-39099R1) entitled ‘’ Barriers and facilitators to physical activity: a comparative analysis of transplant athletes competing in high intensity sporting events with other transplant recipients''.

Overall, this study provides a novel approach. The idea of research is very interesting, well written and reasonable. I would like to thank the authors for their successful work to address the reviewers' comments. The authors have done great efforts to accomplish this work. They fulfilled all comments and made necessary changes throughput the manuscript. I recommend accepting the manuscript its revised form.

Reviewer #3: Please note that I did not review the initial submission. This is an interesting mixed methods study of barriers to and facilitators of physical activity in transplant athletes. The manuscript is generally well written and presents interesting findings, but I note some key points that should be addressed.

1. As per PLOS ONE guidelines, please consider including the COREQ checklist (http://journals.plos.org/plosone/s/submission-guidelines#loc-qualitative-research).

2. In their response to initial reviews, the authors provided a doi for the quantitative data. Unfortunately, the doi did not work for me and I was unable to access the data. I tried searching for ORDO without luck. Please provide the correct link/doi for the data.

3. Regarding the comparisons with data from study by Van Adrichem et al., I see that descriptive data is available for each participant in the Van Adrichem study, but for the other comparisons (e.g., % that reported physical limitations, among others) - did the authors conduct test to compare proportions? Please clarify and modify the methods accordingly. Please also clarify how effect size was determined. Is this Also, as per the paragraph on "Participant selection" in the Van Adrichem study, "the exact level of physical activity was not assessed" so it does not seem appropriate to describe that cohort as having average activity levels.

4. In the last paragraph of the introduction, the authors mention 2 study aims. Yet, the 2nd aim is not mentioned in the abstract, and I did not see any results presented to support this aim.

5. Figure 1 requires modifications as the axis labels do not accurately describe the data presented - is this meant to be the % of TxA who reported barriers/facilitators?

Additional comments

1. Abstract, Aim: Consider modifying the description of the study population ("physically active transplant recipients") to more accurately describe the group (transplant athletes) as is used elsewhere in the manuscript.

2. Abstract, Aim: Use past tense - change analyses to analysed. Similarly change the aims in the introduction to past tense (The first aim of this study was to...").

3. What is meant by "visual artefact"? Please consider including the "visual artefact" in the manuscript or as supplemental material. Perhaps it could be included with the interview guide.

4. Line 63: Change the wording of "...recipients indicated to..."

5. Line 83: Gender refers to Gender refers to the socially constructed roles, behaviours, expressions and identities of girls, women, boys, men, and gender diverse people - please clarify how gender was assessed in this study.

(https://researchintegrityjournal.biomedcentral.com/articles/10.1186/s41073-016-0007-6)

6. Line 160: How did the authors determine the training intensity of the TxA that didn't volunteer for an interview?

7. Line 164: Remove the period after size.

8. Lines 452-453: Please provide references to support the statement about the wider transplant community and the adult population in general.

9. Lines 466-468: Similar to my comment above, data to support the 2nd aim was not presented in the results. Hours of training are mentioned on line 199, but this doesn't appear to be specific to the time of the WTG.

10. Line 484: Change IronMan to Ironman

11. Line 506: Please modify the wording of "...an whether in line..."

7. PLOS authors have the option to publish the peer review history of their article (what does this mean?). If published, this will include your full peer review and any attached files.

Reviewer #2: No

Reviewer #3: No

---

## [Author Response · Author response to Decision Letter 1]

16 Jun 2024

Please see attached revision document

---

## [Decision Letter · Decision Letter 2]

23 Jun 2024

PONE-D-23-39099R2Barriers and facilitators to physical activity: a comparative analysis of transplant athletes competing in high intensity sporting events with other transplant recipients.PLOS ONE

Dear Dr. Duncan,

Thank you for submitting your manuscript to PLOS ONE. After careful consideration, we feel that it has merit but does not fully meet PLOS ONE’s publication criteria as it currently stands. Therefore, we invite you to submit a revised version of the manuscript that addresses the points raised during the review process.

 Thank you for addressing my comments. I noted a few additional points below that require clarification. 

We look forward to receiving your revised manuscript.

Kind regards,

Heather Macdonald, Ph.D

Academic Editor

PLOS ONE

Journal Requirements:

Reviewers' comments:

Reviewer's Responses to Questions

**Comments to the Author**

1. If the authors have adequately addressed your comments raised in a previous round of review and you feel that this manuscript is now acceptable for publication, you may indicate that here to bypass the “Comments to the Author” section, enter your conflict of interest statement in the “Confidential to Editor” section, and submit your "Accept" recommendation.

Reviewer #3: (No Response)

2. Is the manuscript technically sound, and do the data support the conclusions?

Reviewer #3: (No Response)

3. Has the statistical analysis been performed appropriately and rigorously? 

Reviewer #3: (No Response)

4. Have the authors made all data underlying the findings in their manuscript fully available?

Reviewer #3: (No Response)

5. Is the manuscript presented in an intelligible fashion and written in standard English?

Reviewer #3: (No Response)

6. Review Comments to the Author

Reviewer #3: Thank you for responding to my comments. There remain a few issues to be addressed:

1. Unfortunately, the COREQ checklist was not included with the PDF submission. Please also consider referring to all supplementary files as such in the manuscript rather than using the term "Appendix".

2. Apologies for my confusion, but I am still unclear how the authors obtained the data from the Van Adrichem et al. study that was used to conduct the Mann Whitney U tests. The authors provide the data for the TxA in an excel file - do they have a similar file for the Van Adrichem study? Please clarify in the manuscript.

3. If a specific question about gender was not asked, please change gender to sex throughout the manuscript, as male/female are categories used to describe sex not gender.

4. Regarding the training intensity of the TxA that didn't volunteer - please include the information about Strava in the manuscript.

5. Regarding Figure 1 - sorry to harp on this, but I still feel that the figure title could be modified to more clearly reflect what is presented. The figure shows a scatterplot of the proportion of TxA that reported barriers to and facilitators of to PA. I feel it is important to clarify the proportion aspect - it is not a scatterplot of just barriers and facilitators on their own. Also, in the figure, the title at the top is not needed since you have a figure caption. Consider also using different colours or perhaps symbols for the type of transplant. It is not possible to distinguish between the heart and kidney participants, and the bone marrow circles are also difficult to distinguish with the black outline.

7. PLOS authors have the option to publish the peer review history of their article (what does this mean?). If published, this will include your full peer review and any attached files.

Reviewer #3: No

---

## [Author Response · Author response to Decision Letter 2]

27 Jun 2024

Please find attached the requested documents.

---

## [Editor Report · Decision Letter 3]

1 Jul 2024

Barriers and facilitators to physical activity: a comparative analysis of transplant athletes competing in high intensity sporting events with other transplant recipients.

PONE-D-23-39099R3

Dear Dr. Duncan,

We’re pleased to inform you that your manuscript has been judged scientifically suitable for publication and will be formally accepted for publication once it meets all outstanding technical requirements.

Kind regards,

Heather Macdonald, Ph.D

Academic Editor

PLOS ONE
---

## [Editor Report · Acceptance letter]

16 Jul 2024

PONE-D-23-39099R3 

PLOS ONE

Dear Dr. Duncan, 

I'm pleased to inform you that your manuscript has been deemed suitable for publication in PLOS ONE. Congratulations! Your manuscript is now being handed over to our production team.

Kind regards, 

on behalf of

Dr. Heather Macdonald 

Academic Editor

PLOS ONE